# Botulinum Toxin Injection in Children with Hemiplegic Cerebral Palsy: Correction of Growth through Comparison of Treated and Unaffected Limbs

**DOI:** 10.3390/toxins11120688

**Published:** 2019-11-23

**Authors:** You Gyoung Yi, Dae-Hyun Jang, Dongwoo Lee, Ja-Young Oh, Mi-Hyang Han

**Affiliations:** 1Department of Rehabilitation Medicine, Seoul National University Hospital, Seoul National University College of Medicine, Seoul 03080, Korea; lyk861124@gmail.com; 2Department of Rehabilitation Medicine, National Traffic Rehabilitation Hospital, Gyeonggi-do 12564, Korea; 3Department of Rehabilitation Medicine, College of Medicine, The Catholic University of Korea, Seoul 06591, Korea; violetfear1@naver.com (D.L.); arcayje@hanmail.net (J.-Y.O.); gold830@naver.com (M.-H.H.)

**Keywords:** muscle thickness, botulinum toxin, fascicle angle, fascicle length, hemiplegic cerebral palsy

## Abstract

Botulinum toxin type A (BoNT-A) injections in children with cerebral palsy (CP) may negatively affect muscle growth and strength. We injected BoNT-A into the affected limbs of 14 children (4.57 ± 2.28 years) with hemiplegic CP and exhibiting tip-toeing gait on the affected side and investigated the morphological alterations in the medial head of the gastrocnemius muscle (GCM). We assessed thickness of the GCM, fascicle length, and fascicle angle on the affected and unaffected sides at baseline at 4 and 12 weeks after BoNT-A injections. The primary outcome measure was the change (percentage) in GCM thickness in the affected side treated with BoNT-A in comparison with the unaffected side. The percentage of treated GCM thickness became significantly thinner at 4 and 12 weeks after BoNT-A injection than baseline. However, the percentage of fascicle length and angle in treated limbs showed no significant change from baseline 4 and 12 weeks after the injection. BoNT-A injections might reduce muscle thickness in children with spastic hemiplegic CP. Fascicle length and angle might not be affected by BoNT-A injections after correction of normal growth of the children.

## 1. Introduction

In spastic cerebral palsy (CP), muscle strength is usually compromised, resulting in a decrease in functional capacity [1,2]. Botulinum toxin type A (BoNT-A) is often injected into the affected muscles to reduce spasticity and improve gait function in children with CP [2,3,4,5,6]. However, BoNT-A injections in children with CP may negatively affect muscle growth and strength.

Previously studies on the medial head of the gastrocnemius muscle (GCM) in children with spastic CP revealed that muscle volume is 22% smaller than that in typically developing peers between the ages of 2 and 5 years [5] and 37% smaller than that in unaffected young adults [4]. In addition, the fascicle length in the affected calf of patients with hemiplegic CP is shorter than that in the unaffected calf [7]. In contrast, it is reported that the thickness and cross-sectional area of the GCM in ambulatory children with spastic CP did not differ from those in typically developing peers when the lower leg length was corrected [2]. However, GCM muscle thickness and length in the affected side have rarely been compared with those in the unaffected side.

Several studies revealed the structural changes in muscles after BoNT-A injection in spastic CP [5,8,9,10,11,12,13,14,15,16]. In a study of 15 children with spastic ambulatory diplegic CP aged 5–11 years, 5 weeks after BoNT-A injection, GCM volume decreased by 4.47% [12]. It was also reported that the total volume of the treated muscle remained unchanged. It was hypothesized that the GCM volume did not change because of the hypertrophy of the synergistic muscles [12].

GCM volume and length are correlated with age in children with spastic CP and in typically developing peers [5]. Therefore, because children grow over time, if muscle thickness and volume remain the same after toxin injection, structural change has actually occurred, in as much as the ages at the times of evaluation are different. In this study, we investigated the morphological alterations in the GCM after BoNT-A injection in children with spastic hemiplegic CP by comparing the affected limbs treated with BoNT-A and the unaffected limb.

## 2. Results

### 2.1. Study Participants

A total of 14 children with hemiplegic spastic CP were enrolled. Baseline characteristics for the children are presented in Table 1. According to the Gross Motor Function Classification System (GMFCS), all participants were at functional level I or II. Their mean age was 4.57 years (standard deviation, 2.28 years; range, 2–10 years). Table 2 lists the sites of injections and the dose of BoNT-A injected into the medial head of the GCM ranging from 2.61–4.35 U/kg of body weight.

### 2.2. Changes in Medial Head of the Gastrocnemius Muscle Thickness After Injection

In the limbs treated with BoNT-A injection, the thickness of the GCM tended to decrease from the time of injection to 4 weeks later (Table 3) without any statistical significance (*p* > 0.05 for repeated-measures analysis of variance (ANOVA) for −10°, 0°, and 10° ankle angle). However, unaffected GCM thickness increased at 12 weeks in comparison with both baseline and 4 weeks after the injection (Table 3). When evaluated at 10° ankle plantarflexion and dorsiflexion, GCM thickness on the treated side, in comparison with the unaffected side, significantly decreased 4 and 12 weeks after the injection (Table 4). Furthermore, in the neutral ankle position, GCM thickness on the treated side, in comparison with the unaffected side, significantly decreased 12 weeks after the injection (Table 4).

### 2.3. Changes in Fascicle Length After Injection

In the treated side, fascicle length increased 12 weeks after the injection in all three ankle positions (Table 3). However, on the unaffected side, it significantly increased 4 and 12 weeks after injection (Table 3). Fascicle length on the treated side, in comparison with the unaffected side, showed no significant change 4 and 12 weeks after the injection (Table 4).

### 2.4. Changes in Fascicle Angle After Injection

Fascicle angle showed a tendency to decrease on the treated side (Table 3) but not significantly (*p* > 0.05 for repeated-measures ANOVA for −10°, 0°, and 10° ankle angle). On the unaffected side, fascicle angle significantly decreased 4 and 12 weeks after injection (Table 3). Fascicle angle on the treated side, in comparison with the unaffected side, showed no significant change 4 and 12 weeks after the injection (Table 4).

## 3. Discussion

We analyzed the structural changes after BoNT-A injection into muscles affected by hemiplegic CP to correct normal growth of the children. On the unaffected side, GCM thickness and fascicle length increased and fascicle angle decreased 12 weeks after the injection. On the treated side, fascicle length increased 12 weeks after the injection and muscle thickness showed a tendency to decrease four weeks after the injection. GCM thickness on the treated side, in comparison with the unaffected side significantly reduced 12 weeks after the injection.

### 3.1. Changes in Medial Head of the Gastrocnemius Muscle Thickness After Injection

Muscle force-generating capacity is influenced by muscle cross-sectional area [3,4,10,11,17]. Muscle cross-sectional area is highly correlated with muscle thickness, and muscle thickness measured using ultrasonography can be used as a surrogate marker of change in muscle strength [3].

In this study, muscle thickness on the treated side tended to decrease 4 weeks after injection and did not show recovery at 12 weeks. On the unaffected side, muscle thickness significantly increased at 12 weeks. The increase in GCM thickness over time on the unaffected side may be due to the natural growth of children but could also be attributed to the compensatory changes because of contralateral GCM weakness. Compared with the unaffected side, muscle thickness on the treated side decreased at 4 weeks (80.51%–82.69%) and decreased even more at 12 weeks (74.76%–76.09%).

Recently, Multani et al. reported that human volunteers and experimental animals showed muscle atrophy after injections of BoNT-A for at least 12 months [17]. It is unclear whether these changes, mediated at the molecular level, are reversible; muscle atrophy was accompanied with the loss of muscle contraction and replacement by fat and connective tissue [17]. Multani et al. concluded that clinical protocols for using BoNT-A must be revised so that it is used more carefully and less frequently and the effects of muscle injection must be monitored more carefully for short- and long-term benefits and harm [17]. As the number of BoNT-A injections increase in spastic CP, the percentage of type 2 fibers in muscles increases and the percentage of type 1 fibers decreases [8]. In line with the previous studies, our results also suggest that BoNT-A injection might inhibit normal muscle growth for at least 12 weeks, although we did not study the fiber changes at a molecular level.

However, no correlation was observed between the dose injected into the GCM (U/kg) and the percentage of atrophy at 4 and 12 weeks (*p* = 0.928 and 0.513, respectively). There may be a correlation between the injection dose and the percentage of atrophy; however, this correlation was not found to be significant in the present study. This could be because of the small sample size and the administration of similar doses (2.61–4.35 U/kg). Therefore, further studies are necessary to determine the correlation between the injection dose and percent of atrophy.

Although the mechanisms and the clinical factors that may lead to atrophy are poorly studied, muscles have the propensity to accumulate lipids that may lead to an underestimation of atrophy in these muscles following BoNT injections [18]. Despite clinicians being aware of this fact, the literature regarding its presence is limited, and physicians must keep in mind that an underestimation of the actual extent of muscle atrophy is common due to fat deposition. In cases of repetitive injections, it would be suitable to use electromyography- or ultrasonography-guided injections to better localize the bulk of the muscle fiber containing a high level of lipid deposition [18]. In addition, it is necessary to refrain from indiscriminate procedures and to perform strength exercises after the procedure because muscle thickness decreases following BoNT injection.

### 3.2. Changes in Fascicle Length After Injection

Fascicle length is the primary determinant of muscle excursion because it represents the number of sarcomeres working in series [3]. Short muscle fascicles have a reduced number of sarcomeres to decrease the maximum shortening speed [3,5]. The fascicle length in the affected calf of patients with hemiplegic CP is shorter than that in the unaffected calf [7].

In the present study, baseline fascicle length on the affected side was less than that on the unaffected side, which is a finding similar to those in previous studies [7,19,20]. Fascicle length on the treated side significantly increased at 12 weeks, and it significantly increased on the unaffected side as well. In comparison with the unaffected side, fascicle length on the treated side (affected side fascicle length/unaffected side fascicle length×100) did not show any significant change within 12 weeks after the injection. In contrast, Kawano et al. reported an increase in fascicle length of 155.1% after BoNT-A treatment in children with spastic CP (mean age, 6.2 years) [19]. In contrast to this study, in which the fascicle length was measured at a fixed angle, they measured the fascicle length in maximum dorsiflexion angle. It is possible that the natural increase in fascicle length as a result of growth might have not been considered because the changes in fascicle length on both sides were not compared in the previous study. Another reason might be the compensatory change of the GCM on the unaffected side by fascicle lengthening after the weakness of the GCM on the affected side.

### 3.3. Changes in Fascicle Angle After Injection

Fascicle angle is reported to affect muscle force production [3]. In pennate muscles, such as the gastrocnemius, muscle fascicles are obliquely arranged. In these muscles, the fascicle angle, indicating the position of muscle fascicles in relation to the aponeurosis, considerably affects the muscular force during contraction [3,19].

Chen et al. reported that the fascicle angle in children with spastic CP is larger than that in typically developing peers [7]. However, Mohagheghi et al. [21] revealed that fascicle angles in the paretic and nonparetic legs in children with hemiplegic CP were similar. In our study, fascicle angles of both sides at baseline assessments were similar (*p* > 0.05). Kawano et al. reported that the fascicle angle in patients with CP significantly decreased after injection: from 28.2° to 25.8° in the resting position of the ankle and from 18.6° to 15.9° in the maximum dorsiflexion position of the ankle [19]. In the present study, the fascicle angle on the affected side tended to decrease within 12 weeks and that on the unaffected side also decreased over time. The fascicle angles on the two sides were not significantly different 12 weeks after the injection. In Kawano et al.’s study, fascicle angle was measured at resting position and maximal dorsiflexion angle; our measurement method was different. It is possible that the fascicle angle decreased over time as a result of the normal growing process. Moreover, it might be attributable to the compensatory change after contralateral side weakness. Our findings indicate that the tendency of the fascicle angle on the treated side to decrease might not be affected by BoNT-A injection. Therefore, as the fascicle length and angle in the affected limb did not differ from those in the unaffected limb, the decrease in GCM thickness in the affected limb could indicate the pathological condition of CP after BoNT-A injection.

### 3.4. Study Limitations

The study has several limitations. First, this was a single-center study with a small number of subjects. Second, the affected and unaffected GCMs may not grow at the same rate in patients with hemiplegic CP even if BoNT-A injection is not performed. Moreover, because the study subjects included only children with GMFCS level I and II function who could walk, the results for children with higher GMFCS levels or in the upper limb muscles may differ. Because this study only included ambulatory children with GMFCS levels I and II, most did not show a dramatic functional improvement after the injection. Except for one child, every child showed a reduction in one level according to the modified Ashworth scale; therefore, analyzing the correlation of change according to the modified Ashworth scale with the degree of atrophy or dose was impossible. Lastly, it is difficult to presume that the structural change was caused only by BoNT-A because the conventional rehabilitation program was performed after BoNT-A injection. Although active movement training with a trained physical therapist was also performed, stretching exercises might have influenced the fascicle length and angle. However, non-performance rehabilitation therapy after the procedure was difficult in the clinical situation.

## 4. Conclusions

BoNT-A injections in the medial head of the GCM might reduce muscle thickness in children with spastic hemiplegic CP for up to 12 weeks after the injection. Therefore, clinicians should refrain from indiscriminate procedures and perform strength exercises after BoNT-A injection. However, fascicle length and angle in the treated affected GCM showed no significant change in comparison with the unaffected side.

## 5. Materials and Methods

### 5.1. Participants

We included children with spastic hemiplegic CP aged 2–18 years and had tip-toeing gait on the affected side. We excluded children (1) who had an allergic reaction to BoNT-A; (2) who, during screening, were participating in other clinical trials or had taken muscle relaxants, benzodiazepine, or anticholinergic drugs within 4 weeks of screening; (3) who had received a BoNT-A injection within the previous six months; and (4) who had been treated with intrathecal baclofen, selective dorsal rhizotomy, or other orthopedic surgery.

This study was registered as a clinical trial (number KCT0004018). All procedures performed in the study were in accordance with the ethical standards of the institutional or national research committee and with the 1964 Helsinki declaration and its later amendments or comparable ethical standards. All participants provided written informed consent before their participation and the Incheon St. Mary’s Institutional Review Board approved the study design (No. OC16OISI0017).

### 5.2. Procedure

For each patient, one vial-A (100 U) of onabotulinum toxin A (Botox^®^; Allergan, Inc., Irvine, CA, USA) was diluted in 2 mL of normal saline to a concentration of 5 U per 0.1 mL. The maximum dose provided was 200 U. The dose for each muscle to be injected was calculated before the injection. The injections were delivered via an injectable monopolar needle electrode (37 mm; Chalgren, Gilroy, SK, Canada) guided by electromyography, by a physician who had at least 5 years of experience in administering BoNT-A injections for children with CP. The dose injected into the GCM ranged from 2.61–4.35 U/kg of body weight. The sites of injection are listed in Table 2. All participants underwent a conventional rehabilitation program of combined passive ankle stretching and active movement training with a trained physical therapist thrice per week for 12 weeks. No additional strengthening program or intensive treatment was prescribed for the participants. None of the participants had additional serial casting after injection.

### 5.3. Assessment

Ultrasound images of GCMs were taken by physicians using a HD11 XE ultrasound system (Philips, Bothell, WA, USA) with a scanning frequency of 7–12 MHz and a line-array transducer. The ultrasound image of the GCM was obtained in accordance with the recommendations of Barber et al. [4]. The injection into GCM was made approximately at 25% of the distance from the medial malleolus to the popliteal fossa.

All ultrasound measurements were obtained with the ankle in 10° of plantarflexion, at a neutral angle (0°), and at 10° of dorsiflexion. GCM thickness, fascicle length, and fascicle angle were measured as shown in Figure 1. Muscle thickness (Figure 1A) was defined as the longest distance between fascia of GCM in the cross-sectional area; the fascicle length (Figure 1B) was calculated as the linear distance between the insertion of a fascicle into the lower and upper aponeuroses; and fascicle angle (Figure 1C) was defined as the angle between muscle fiber and estimated parallel fascia line.

The primary outcome measure was the change (percentage) in GCM thickness in the affected side treated with BoNT-A in comparison with unaffected side at baseline, 4, and 12 weeks after injection. The secondary outcomes were the changes in GCM fascicle length and angle on the affected side treated with BoNT-A in comparison with unaffected side before injection and at 4 and 12 weeks after injection. All postinjection adverse events were documented. Discomfort, reduced general condition, headache, dry mouth, injection site pain, viral infection, dizziness, back pain, and tiredness were also investigated and documented.

### 5.4. Statistical Analysis

Statistical analyses were performed using SPSS 22 software (IBM Corporation, Armonk, NY, USA). A *p*-value of 0.05 was used as the threshold for statistical significance. Repeated-measures ANOVA was conducted to examine the changes in muscle thickness, fascicle length, and fascicle angle.

## Figures and Tables

**Figure 1 toxins-11-00688-f001:**
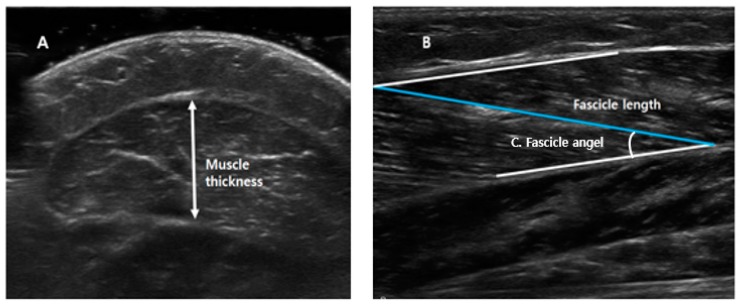
Measurements of medial head of the gastrocnemius muscle thickness, fascicle length, and fascicle angle. (**A**) Muscle thickness: the longest distance between fascia of gastrocnemius muscle in cross-sectional area. (**B**) Fascicle length: the linear distance between the insertion of a fascicle into the lower and upper aponeuroses. (**C**) Fascicle angle: the angle between muscle fiber and estimated parallel fascia line.

**Table 1 toxins-11-00688-t001:** Baseline characteristics of the study participants.

Characteristics	Total (*N* = 14)
Sex (*n*)	
Male	10 (71.43%)
Female	4 (28.57%)
Age (mean), years	4.57 (*SD* = 2.28)
Body weight (mean), kg	21.17 (*SD* = 8.24)
History of previous BoNT-A injection *(n)*	9 (64.29%)
GMFCS level, *(n)*	
I	12 (85.71%)
II	2 (14.29%)
Type *(n)*	
Right hemiplegia	9 (64.29%)
Left hemiplegia	5 (35.71%)

**Table 2 toxins-11-00688-t002:** Site of botulinum toxin A injection in children with hemiplegic spastic cerebral palsy.

Location of injection *	*n*
Medial head of gastrocnemius muscle ^†^	14 (100%)
Lateral head of gastrocnemius muscle	8 (57.14%)
Triceps posterior muscle	6 (42.86%)
Peroneus longus muscle	2 (14.29%)
Soleus muscle	1 (7.14%)
Upper extremity muscles	3 (21.43%)

* Total dose of injection ranged from 5.08–14.94 U/kg of body weight. ^†^ Injected dose into the medial head of gastrocnemius muscle ranged from 2.61–4.35 U/kg of body weight.

**Table 3 toxins-11-00688-t003:** Change in the medial head of the gastrocnemius muscle thickness, fascicle length, and fascicle angle: comparison of the treated side with the unaffected one (N = 14).

	Toxin Injected Affected Side, Mean (*SD*)			Unaffected Side, Mean (*SD*)		
	−10°	0°	10°	−10°	0°	10°
Baseline MT (cm)	1.33 (0.22)	1.34 (0.21)	1.35 (0.21)	1.51 (0.24)	1.49 (0.24)	1.50 (0.26)
After 4 weeks MT (cm)	1.24 (0.18)	1.26 (0.20)	1.25 (0.21)	1.54 (0.22)	1.52 (0.22)	1.54 (0.22)
After 12 weeks MT (cm)	1.25 (0.27)	1.26 (0.27)	1.27 (0.29)	1.69 (0.27) ^†,††^	1.67 (0.24) ^†,††^	1.68 (0.22) ^†,††^
Baseline FL (cm)	4.36 (0.79)	4.68 (0.87)	4.98 (0.96)	4.97 (0.89)	5.21 (0.88)	5.66 (1.06)
After 4 weeks FL (cm)	4.60 (0.92)	4.95 (0.99) *	5.29 (1.09)	5.22 (1.13) *	5.57 (1.14) *	6.03 (1.19) *
After 12 weeks FL (cm)	4.79 (0.95) ^†^	5.17 (1.07) ^†^	5.50 (1.04) ^†^	5.66 (1.17) ^†,††^	5.93 (1.06) ^†,††^	6.32 (1.15) ^†,††^
Baseline FA (°)	14.03 (1.59)	13.24 (1.61)	12.40 (1.47)	15.39 (2.60)	14.52 (2.12)	13.58 (1.81)
After 4 weeks FA (°)	13.62 (1.90)	12.50 (1.70)	11.79 (1.99)	13.88 (2.54) *	12.87 (2.12) *	12.00 (1.64) *
After 12 weeks FA (°)	12.85 (1.53)	11.61 (1.07) ^†^	11.46 (1.02)	12.99 (2.18) ^†^	12.34 (2.07) ^†^	11.40 (1.18) ^†^

FA, fascicle angle; FL, fascicle length; MT, muscle thickness; SD, standard deviation. −10° indicates 10° ankle plantarflexion; 0° indicates neutral ankle angle; 10° indicates 10° ankle dorsiflexion. state. * Interval from baseline to 4 weeks after injection. ^†^ Interval from baseline to 12 weeks after injection. ^††^ Interval from 4–12 weeks after injection. Bonferroni adjusted *p* < 0.05.

**Table 4 toxins-11-00688-t004:** Percentage of muscle thickness, fascicle length, and fascicle angle of the treated limb compared with the unaffected one.

Muscle architecture	−10°	0°	10°
Muscle thickness (%) Baseline, mean (SD)	88.38 (7.80)	89.84 (6.37)	90.53 (6.92)
After 4 weeks, mean (SD)	80.51 (7.87) *	82.69 (7.63)	81.41 (8.58) *
After 12 weeks, mean (SD)	74.76 (15.03) ^†^	76.09 (15.04) ^†^	75.62 (15.50) ^†^
Fascicle length (%) Baseline, mean (SD)	88.88 (8.04)	89.85 (7.85)	87.96 (6.11)
After 4 weeks, mean (SD)	88.81 (11.78)	89.36 (9.77)	87.91 (9.77)
After 12 weeks, mean (SD)	84.88 (7.87)	86.93 (7.36)	87.05 (7.40)
Fascicle angle (%) Baseline, mean (SD)	93.90 (21.45)	92.56 (15.83)	92.03 (11.20)
After 4 weeks, mean (SD)	99.89 (16.83)	98.51 (14.85)	99.37 (17.98)
After 12 weeks, mean (SD)	100.40 (13.22)	95.87 (13.47)	103.36 (18.30)

−10° indicates 10° ankle plantarflexion; 0° indicates neutral ankle angle; 10° indicates 10° ankle dorsiflexion. * Interval from baseline to 4 weeks after injection. ^†^ Interval from baseline to 12 weeks after injection. Significant changes in interval from 4–12 weeks after injection were not present. Bonferroni adjusted *p* < 0.05. SD, standard deviation.

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
