# Peer review of "Botulinum Toxin Injection in Children with Hemiplegic Cerebral Palsy: Correction of Growth through Comparison of Treated and Unaffected Limbs"

_toxins, 2019, doi:10.3390/toxins11120688_

Round 1

Reviewer 1 Report

Thank you for giving me this great opportunity to review the present
interesting study.

This is a very carefully designed study.
Many statistical tests are also appropriate.

In comparison between the affected limb and the unaffected limb, why did the GCM thickness, fascicle length, and angle changed in the unaffected limb?These changes seemed to be statistically significant. Is this a compensatory change due to Botulinus toxin injection on the affected side?

The muscle changes of unaffected side seem to be treated as a control group, but if they are statistically significant changes, they cannot be overlooked. Therefore, the author need to discuss in Discussion.

This study performed rehabilitation therapy. Although it is unclear whether previous studies have performed rehabilitation, the fact that the fascicle length and fascicle angle in the affected side were not significantly different from those on the unaffected side may be the effect of appropriate stretch. Muscle strengthening is a difficult problem in CP patients.

Therefore, in the situation where the fascicle length and angle in the affected limb are not different from those in the unaffected limb side, the decrease in GCM thickness in the affected limb is considered to indicate the pathological condition of CP.

If there are additional discussion accompanying the above, this study will become significance and future prospect of this research.

I was worried that the same expression appeared in Introduction and Discussion. this indicate poor as article.

Conclusion was not found.

Author Response

We appreciate your critical and useful comments. Changes to the text are given in red text in the revised manuscript.

Point 1: In comparison between the affected limb and the unaffected limb, why did the GCM thickness, fascicle length, and angle changed in the unaffected limb? These changes seemed to be statistically significant. Is this a compensatory change due to Botulinus toxin injection on the affected side?

Response 1:

We think the reason that the GCM thickness, fascicle length, and angle changed in the unaffected limb is owing to natural growth of children. Also, in part, we think it might be a compensatory change as you suggested. We have added these findings in the discussion, and we used the ratios (affected/unaffected) as outcome measure.

3.1. Changes in medial head of the gastrocnemius muscle thickness after injection

The increase in GCM thickness over time on the unaffected side may be due to the natural growth of children but could also be attributed to the compensatory changes because of contralateral GCM weakness.

3.2. Changes in fascicle length after injection

It is possible that the fascicle angle decreased over time as a result of the normal growing process. Moreover, it might be attributable to the compensatory change after contralateral side weakness. Our findings indicate that the tendency of the fascicle angle on the treated side to decrease might not be affected by BoNT-A injection. Therefore, as the fascicle length and angle in the affected limb did not differ from those in the unaffected limb, the decrease in GCM thickness in the affected limb could indicate the pathological condition of CP after BoNT-A injection.

Point 2: The muscle changes of unaffected side seem to be treated as a control group, but if they are statistically significant changes, they cannot be overlooked. Therefore, the author need to discuss in Discussion.

Response 2: Although we thought it was due to natural growth, at the beginning of the procedure, children could use more of the unaffected leg by compensation as you pointed out. We have further added the possibility in Discussion. Please see response 1.

Point 3: This study performed rehabilitation therapy. Although it is unclear whether previous studies have performed rehabilitation, the fact that the fascicle length and fascicle angle in the affected side were not significantly different from those on the unaffected side may be the effect of appropriate stretch. Muscle strengthening is a difficult problem in CP patients.

Response 3: We totally agree with your critical comment. It is unclear the fact that fascicle length and fascicle angle in the affected side were not significantly different from those on the unaffected side may be the effect of appropriate stretch. However, not performing rehabilitation therapy after the procedure is very difficult in the clinical situation. We have added these discussions and limitations in the Discussion. Muscle strengthening in children with cerebral palsy is a difficult problem, and stretching exercises, which can be applied relatively easily, may have influenced the results of this study.

3.4. Study limitations

Lastly, it is difficult to presume that the structural change was caused only by BoNT-A because the conventional rehabilitation program was performed after BoNT-A injection. Although active movement training with a trained physical therapist was also performed, stretching exercises might have influenced the fascicle length and angle. However, non-performance rehabilitation therapy after the procedure was difficult in the clinical situation.

Point 4: Therefore, in the situation where the fascicle length and angle in the affected limb are not different from those in the unaffected limb side, the decrease in GCM thickness in the affected limb is considered to indicate the pathological condition of CP.

Response 4: We have added your comments in discussion. Thank you for your kind suggestion. If there are additional discussion accompanying the above, this study will become significance and future prospect of this research.

3.3. Changes in fascicle angle after injection

Therefore, as the fascicle length and angle in the affected limb did not differ from those in the unaffected limb, the decrease in GCM thickness in the affected limb could indicate the pathological condition of CP after BoNT-A injection.

Point 5: I was worried that the same expression appeared in Introduction and Discussion. this indicate poor as article.

Response 5: We have modified the sentences as you suggested.

Point 6: Conclusion was not found.

Response 6: We have added the conclusion as you pointed out.

Conclusions

BoNT-A injections in the medial head of the GCM might reduce muscle thickness in children with spastic hemiplegic CP for up to 12 weeks after the injection. Therefore, clinicians should refrain from indiscriminate procedures and perform strength exercises after BoNT-A injection. However, fascicle length and angle in the treated affected GCM showed no significant change in comparison with the unaffected side.

Reviewer 2 Report

The figure 2 (A, B, D, E, H) and table 3 show basically the same data. I suggest keeping only data in the table 3, or the graph with SD and statistics, not both. Regarding the table 4, it shows exactly the same data as the table no 3. Was this erroneously submitted? Shouldn't the table 4 contain the similar data as the figure 2 (C, F, I) ?

Is the % of atrophy connected to the dose injected into the gastrocnemius and/or total dose injected in individual patients? It would be interesting to calculate for possible correlation and to include the graph of dose vs. percentage of atrophy (baseline vs 4 or baseline vs 12 wk).

The text does not report the functional outcome of the treatment. Have the measures of improvement been assessed in these patients (e.g. change in modified Ashworth scale etc.). It would be interesting to correlate also the degree of atrophy or dose with functional outcome of the treatment.   

The authors do not discuss the possible long-term deleterious outcomes of muscle changes, which could be of interest to the physicians and non-expert readers. I suggest an additional paragraph of the text related to this matter.

Author Response

Please see attatched files (due to many figures)

Reviewer 3 Report

The authors state that BoNT-A injections might reduce muscle thickness in children with spastic hemiplegic CP. Fascicle length and fascicle angle might not affected by BoNT-A injections after correction of normal growth of the children.

This is an intersting result, however, it is not clear to the reader what the clinical implication is. How does this influence our clinical practice, what can we learn from that finding? Can this result be transfered to other muscles, i.e. on the forearm?

The paper might be resubmittted aftre major changes.

Round 2

Reviewer 3 Report

The paper can be accepted if space is available.